# Septic Transfusion Reactions Involving *Burkholderia cepacia* Complex: A Review

**DOI:** 10.3390/microorganisms12010040

**Published:** 2023-12-25

**Authors:** Margarita Salamanca-Pachon, Nohora Isabel Guayacan-Fuquene, Michel-Andres Garcia-Otalora

**Affiliations:** 1Blood Bank Sociedad de Cirugía de Bogotá, Hospital de San José, Bogotá 111411, Colombia; coordinacionbancodesangre@hospitaldesanjose.org.co; 2Public Health Research Group, School of Medicine and Health Science, Universidad del Rosario, Bogotá 111221, Colombia; michel.garcia@urosario.edu.co

**Keywords:** sepsis, transfusion, blood component, *Burkholderia cepacia*, adverse event

## Abstract

This review was conducted to assess the global incidence of transfusion-transmitted infections (TTIs) caused by contamination of blood components with the *Burkholderia cepacia* complex (Bcc). Our search encompassed various specialized databases such as Medline/PubMed, Web of Science, Scopus, Scielo, ScienceDirect, and ClinicalKey. An analysis of the literature revealed a total of eleven reported cases where blood components contaminated with Bcc had been transfused, resulting in sepsis among the affected patients. Of these cases, eight were documented in the literature, while the remaining three occurred within the institution involving the authors of this review. A comparative examination was conducted, considering factors such as primary diagnosis, transfused blood component, time elapsed between transfusion and manifestation of symptoms, administration of antibiotics, and final outcome. Interestingly, regardless of the storage temperature, all blood components were found to be susceptible to Bcc contamination. Furthermore, the cases investigated revealed diverse sources of contamination, and it was observed that all the affected patients had compromised immune systems due to underlying illnesses. Based on these findings, a series of preventive strategies were derived to mitigate and decrease the occurrence of similar cases.

## 1. Introduction

The *Burkholderia cepacia* complex (Bcc), previously referred to as *Pseudomonas cepacia*, comprises 24 opportunistic pathogenic species associated with nosocomial infections [1,2]. The Bcc are aerobic Gram-negative bacilli responsible for infections, particularly among patients with cystic fibrosis, chronic granulomatous disease [3], and immunosuppressed individuals [2]. In 1949, the pioneering work of Walter H. Burkholder led to the identification of this group of microorganisms as the causal agent of onion rot [4]. It was subsequently recognized as a human pathogen in the 1950s [1]. The Bcc ranges in size from 1 to 5 um in length and 0.5 to 1 um in width. These non-spore-forming organisms exhibit motility attributed to the presence of one or more flagella. They produce catalase and do not ferment glucose [5]. The Bcc is recognized for its ability to generate various secreted substances, including proteases, lipases, cytotoxins, and hemolysins. These products are well-documented for their virulence and play a role in triggering a potent inflammatory response that can lead to host cell death [1].

Techniques such as recA species-specific polymerase chain reaction and matrix-assisted laser desorption/ionization–time of flight mass spectrometry are used to accurately identify Bcc species [6]. Additionally, the multilocus sequence typing scheme has proven indispensable in identifying previously misclassified Bcc strains and discovering novel species [1]. Selective media like McConkey, Columbia Media Agar, *Burkholderia cepacia* selective agar, Becton–Dickinson *cepacia* medium, oxidation polymyxin lactose agar, and *Pseudomonas cepacia* agar are commonly utilized to support the growth of most Bcc species. Typically, the incubation period ranges from 48 to 72 h at temperatures between 30 and 37 °C, although some species may exhibit optimal growth at 42 °C. It is worth noting that a characteristic odor and color is often associated with Bcc growth [4,7].

There are nine genomovars (distinct genomic groups that are sufficiently different to be classified as different species, but with phenotypes that do not show sufficient robust differences for discriminating them) within the *Bcc* group, each with unique characteristics [1]: *Burkholderia cepacia* (genomovar I), *Burkholderia multivorans* (genomovar II), *Burkholderia cenocepacia* (genomovar III), *Burkholderia stabilis* (genomovar IV), *Burkholderia vietnamiensis* (genomovar V), *Burkholderia dolosa* (genomovar VI), *Burkholderia ambifaria* (genomovar VII), *Burkholderia anthina* (genomovar VIII), and *Burkholderia pyrrocinia* (genomovar IX) [1]. Among these genomovars, *Burkholderia cenocepacia* is widely recognized as the most virulent within the Bcc, as it is associated with a mortality rate five times higher compared with the other genomovars. The identification and differentiation of these genomovars have greatly contributed to the understanding of the pathogenicity and clinical significance of various *Burkholderia* species, enabling more targeted approaches in the diagnosis, treatment, and management of infections caused by these organisms [1,2].

The Bcc can be found in various natural and artificial environments [4]. It has demonstrated a remarkable ability to colonize surfaces such as catheters, intravenous sets, nebulizers, respiratory devices, and even disinfectants, making it particularly concerning during hospital outbreaks [8]. The Bcc poses a significant threat to hospitalized patients due to its resistance to antibiotics, leading to increased morbidity and mortality [4]. Although community-acquired infections are relatively rare, the Bcc exhibits low virulence in healthy individuals [1]. Natural infection with the Bcc occurs through subcutaneous inoculation, ingestion, or inhalation of the pathogen. The ensuing signs and symptoms in affected patients tend to be diverse and nonspecific, primarily influenced by the mode of inoculation and the patient’s immune system status [1].

The respiratory tract and direct inoculation through venous lines are considered the primary entry points for the bacterium into the bloodstream. The Bcc thrives in moist environments and can sustain itself under minimal nutritional conditions, enabling it to colonize ventilation systems and subsequently encounter epithelial barriers. After surviving within epithelial cells and macrophages, it skillfully breaches epithelial barriers and invades underlying tissues, aided by the mobility conferred by its polar flagellum. This relentless invasion eventually culminates in bloodstream infiltration, leading to systemic dissemination throughout the body [1,2]. Bcc infections present significant challenges in their treatment, primarily due to their remarkable resistance to multiple antibiotics [9], including aminoglycosides, polymyxins, quinolones, trimethoprim, chloramphenicol, and host antimicrobial peptides, with notable variations among species.

The Bcc can form biofilms and has the capability to establish both intracellular and chronic infections within the host. Currently, there are no commercially available vaccines against any member of the Bcc [10]. In the context of oncohematology units, especially those with acute myeloid leukemia, without cystic fibrosis [11], outbreaks caused by the Bcc are frequent, often stemming from a single contaminated source, such as antiseptic, intravenous and nebulizer solutions, or medical devices [12]. However, sometimes there is not an obvious environmental source [11].

Blood transfusion is a widely used therapeutic strategy in clinical practice [13]. However, it is important to acknowledge that along with its benefits, there are potential risks involved [14]. Transfusion-transmitted infections (ITTs) encompass a range of diseases that can be transmitted through the transfusion of whole blood or blood components. These infections can be caused by bacteria, viruses, parasites, fungi, and prions [15]. The most common TTI is sepsis, which can result from the transfusion of blood components contaminated with bacteria [16]. Platelet components (PCs) are stored at room temperature (20–24 °C), providing an ideal environment for bacterial growth. Previous aerobic culture surveillance studies have revealed that about 1 in 1000–3000 PCs were found to be bacterially contaminated [17]. However, the actual rates of septic transfusions were much lower, approximately 1 in 25,000 PCs (ranging from 1 in 13,000 to 100,000) [16]. The reported incidence of contamination varies widely based on location [18], collection type (e.g., apheresis vs. whole blood derived), and the level of implementation of mitigation measures. In contrast, the rate of bacterially contaminated red blood cell units is significantly lower, approximately 1 per 2 million transfused red blood cell units [19]. This difference can be primarily attributed to the refrigerated storage of red blood cell units, which hinders the growth of most transfusion-associated bacterial species [19,20].

Extensive research has been conducted on bacterial growth in blood products using in vitro studies [16]. The behavior of this growth is influenced not only by the type of bacteria [19,20,21] but also by factors such as the donor [22,23], the type of matrix (PCs or red blood cells) [19,20,21], and the temperature [24]. Crucial factors determining the occurrence of transfusion-transmitted bacterial infections include leukopenia in patients and a bacterial concentration exceeding 1 × 10^5^ CFU/mL [25]. Moreover, active surveillance has revealed a higher detection rate of bacterially contaminated PCs and septic reactions compared with passive surveillance [26].

Bacteria most commonly infiltrate collected blood during phlebotomy, where the needle can become contaminated with skin commensals and other contaminants during the venipuncture process [16]. While this is the primary route of contamination, there are more infrequent occurrences where the contamination arises from asymptomatic donor bacteremia, originating from various sources like the upper respiratory tract, oral flora, gastrointestinal tract, or female genital tract [16]. Additionally, there is a rare possibility of contamination from the environment due to imperceptible damage to the collection bag [27].

Blood services around the globe have adopted measures to minimize the potential transmission of bacteria through transfusions [16]. These interventions encompass three main approaches: (a) ensuring the introduction of bacteria is prevented during blood donation by using rigorous donor selection, effective skin disinfection, diverting the initial blood volume, and closely monitoring the component production process; (b) using diagnostic techniques, such as component culture and/or rapid detection methods, to identify any presence of bacteria in the blood components, and (c) implementing pathogen reduction techniques.

Although the primary contaminating bacteria of PCs [20] and red blood cells [19] have been characterized, the focus of this review was to specifically identify cases of sepsis associated with the transfusion of blood components contaminated with the Bcc.

## 2. Materials and Methods

### 2.1. PICO Question and Search Strategy

What does the scientific literature reveal about the number of cases and the characteristics of patients who have reported sepsis-type transfusion reactions after being transfused with blood components contaminated with the Bcc? To answer this, a review of the scientific literature was conducted following PRISMA (Preferred Reporting Items for Systematic Reviews and Meta-analyses) guidelines [28]. Several specialized databases were consulted, including Medline/PubMed, Web of Science, Scopus, Scielo, ScienceDirect, and ClinicalKey. The MeSH terms utilized in the search were “Burkholderia*” or “cepacia*” or “Burkholderia cepacia” or “Pseudomonas cepacia” linked to “septic transfusion reaction” or “transfusion adverse event” or “transfusion*”. To construct the search equations, Boolean characters (AND and OR) were used, and the search was restricted to the title and abstract fields (Table 1). The article search was performed between 25 April 2023 and 25 May 2023. Furthermore, the terms were systematically searched in two additional sources: the “Fatalities Reported to FDA Following Blood Collection and Transfusion: Annual Summary for Fiscal Years 2013–2020” and the “SERIOUS HAZARDS OF TRANSFUSION” annual reports spanning from 1998 to 2022. This thorough search strategy was used to ensure comprehensive coverage and to gather relevant data on the topic.

### 2.2. Study Inclusion/Exclusion Criteria

Studies were included in this review if they (i) were original works; (ii) were performed on humans; (iii) related to transfusion; (iv) compared people with and without antibiotics; (v) included an assessment of time since the appearance of adverse transfusion event; and (vi) were written in English, Spanish, French, German, or Japanese. No filter was applied for the publication date of the articles. The exclusion criteria included articles with cases of patients with infections not associated with the Bcc and documents related to sepsis but not associated with the transfusion of blood components. The remaining documents were used as references for the Introduction, Discussion, and Conclusions sections.

### 2.3. Quality Assessment and Risk of Bias

A comprehensive evaluation of the methodological quality of the studies included in the present review was conducted following the Cochrane recommendations. Data analysis and synthesis were performed using the number of cases of patients transfused with Bcc-contaminated blood components as the unit of analysis. Regarding selection bias, the authors used random and stratified identification of cases, effectively ensuring a consistent number and characteristics of clinical cases. Demographic factors, primary diagnosis, signs and symptoms during/after the transfusion, and treatment after the diagnosis of sepsis showed no significant differences among the studies. Although almost all studies described the blinded generation of randomization sequences, an exception was noted in the study by Shastri, J.S. et al. [4], where the method was not explained. Attrition bias was addressed by excluding individuals not directly involved in the transfusion of blood components, and they were associated with patients diagnosed with cystic fibrosis. Regarding reporting bias, it was challenging to locate protocols for all cases to verify the existence of notification bias and thus, it was classified as an uncertain risk. The Cochrane recommendations identified other potential biases, which were particularly relevant for cases found in a specific clinical setting. Overall, this critical analysis of the methodological quality of the studies provided valuable insights into the risks of bias and limitations within the current body of research on patients transfused with Bcc-contaminated blood components.

A table was built to collect general information about the publication (title, authors, journal and year of publication, and objective) and characteristics of the population studied (sex, age, primary diagnosis, signs and symptoms during/after transfusion, time elapsed between transfusion and onset of symptoms, storage time of the blood component before transfusion, transfused blood component, microbiological culture of the patient and blood units, source of contamination, treatment post sepsis diagnosis, and outcome).

## 3. Results

The search process yielded 1675 records, from which 17 relevant studies were selected for this review. A total of 42 documents were used as references in this manuscript, with four of them specifically related to sepsis-type TTI associated with the Bcc and transfusion reactions in the patients involved (Figure 1).

During the search process in different databases using corresponding equations, a total of 1675 records consisting of case reports, articles, and abstracts were initially found. After filtering out duplicates (479) and records that did not meet the search criteria (890), 276 documents remained for screening, out of which 226 were excluded. After requesting 50 studies for review, 20 of them could not be retrieved, resulting in 30 studies that were evaluated. From this evaluation, 17 studies were identified that met the eligibility criteria for this review.

For the composition of this manuscript, 42 full-text documents were referenced, including articles, case reports, abstracts, and hemovigilance reports. Among these documents, only four case reports were found that were related to sepsis-type TTI associated with the Bcc causing transfusion reactions in the involved patients. A total of eleven cases involving Bcc contamination of blood components were identified, leading to sepsis in the affected patients (Table 2). Among these cases, eight were reported in the scientific literature referenced (cases 1–7 and 11), while the remaining three (cases 8–10) were reported by the authors of this manuscript. These three cases involved sepsis-type transfusion reactions in patients who received PCs obtained using apheresis at the Sociedad de Cirugía de Bogotá-Hospital de San José in Bogotá, Colombia (see Table 2). The eight cases identified were classified as studies with a low risk of bias. A summary of the percentage risk for each considered bias is presented in Figure 2. Notably, the main biases observed across all the cases of patients transfused with Bcc-contaminated blood components were related to performance and detection. This stemmed from the lack of masking concerning the type of adverse reaction to the transfusion, affecting the evaluators of the outcome variable, and ultimately influencing the search results in a favorable and exact manner. Performance and detection bias were evident since none of the four selected studies used blinding measures in their descriptions of cases.

In 1979, Rhame et al. reported three cases (our cases 1–3) of nosocomial infection caused by *Pseudomonas cepacia* [29]. Among these cases, two patients developed septicemia, and one patient experienced a mediastinal wound infection. All three patients had received pool transfusions of cryoprecipitate. The investigation revealed that the source of contamination was the water used in the serological bath for thawing the cryoprecipitates. Upon analyzing the events, it was discovered that the water culture showed the growth of *Pseudomonas cepacia*, which subsequently contaminated the entry ports of the blood components.

In 1996, Dinse et al. reported a case of postoperative complications in a 58-year-old female patient following an uncomplicated vaginal hysterectomy [30] (our case 4). The patient developed signs of disseminated intravascular coagulation and fever, which eventually progressed to septic shock. During and after the operation, she received two units of autologous blood. Subsequent blood culture analysis revealed abundant growth of *Pseudomonas cepacia* and *Serratia marcescens* in both the patient’s blood sample and one unit of the autologous blood within 24 h. Additionally, the culture from this blood bag showed slight growth of Staphylococcus epidermidis within 48 h. The second blood unit displayed slight growth of *Cutibacterium acnes*. Despite rigorous investigation, the source of bacterial contamination could not be identified.

In 2002, García-Erce et al. reported three cases (our cases 5–7) of transfusion-transmitted sepsis caused by *B. cepacia*. The contamination source was traced back to the disinfection solution used for donor arm cleansing prior to phlebotomy [8]. Surprisingly, two out of the three patients who experienced this transfusion-transmitted bacterial infection (TTBI) had not been reported to the transfusion medicine department [8]. Subsequently, a retrospective investigation was initiated after the initial patient developed sepsis, revealing that all affected units were collected on the same day in the hospital’s blood bank donation room. During the investigation, it was discovered that a chlorhexidine solution, specifically a 0·5% aqueous solution, was the culprit behind the outbreak. Both unopened bottles of the same solution in the blood bank and pharmacy stocks were found to be contaminated with *B. cepacia*. The chlorhexidine solution responsible for the contamination was manufactured by an external pharmacology laboratory. Notably, conventional microbiological studies revealed that all isolates belonged to the same *B. cepacia* strain, strongly suggesting a manufacturing-related origin of the contamination [8].

In 2020, the Sociedad de Cirugía de Bogotá-Hospital de San José in Bogotá, Colombia, reported three patients that experienced septic shock after receiving transfusions of PCs contaminated with the Bcc (our cases 8–10). The cases were promptly reported to the National Hemovigilance Program, which since 2016 adopted the definitions of Adverse Transfusion Reactions [31]. The PCs were obtained using apheresis from a single 38-year-old male donor. The imputability of the contamination was definitively established, and the severity of the cases was classified as grade 3. All three patients succumbed at different intervals after the transfusion: 157, 70, and 21 days, respectively. However, sepsis was ruled out as the direct cause of death. Despite an exhaustive investigation, the primary source of contamination remained elusive. Neither the donor, the apheresis kit, the shakers, nor the storage environment of the PCs revealed any traces of the causative agent. Nevertheless, the isolation of the same pathogen in all three contaminated PCs, as well as in the transfused patients, strongly suggests that the contamination might have occurred within the bags containing the PCs. This could have been due to either *B. cepacia* contamination during the manufacturing process of the PCs or the presence of unnoticed damage in the main bag during the handling of blood components throughout the transfusion chain. Notably, these findings align with recent research proposed by Ramirez-Arcos et al. [32].

De et al. reported a 57-year-old male diagnosed with acute myeloid leukemia who experienced septicemia caused by *B. cepacia* [4], which was the number the eleventh case of this review. However, the source of the infection was not specified. During his medical treatment, the patient received PC and red blood cell transfusions. Unfortunately, the specific treatment and outcome details were not provided in the available information. Importantly, the possibility of the infection being associated with the transfusions could not be ruled out.

Finally, we conducted a comparison of TTI rates by analyzing hemovigilance reports from multiple countries [16,17,18]. The data analysis incorporated medians and interquartile ranges, providing valuable statistical measures that captured the variability in the data and enhanced our understanding of infection rates (Figure 3). We estimated that there is a 2412-fold higher chance of identifying a unit of PC contaminated with bacteria compared to detecting HTLV contamination. Similarly, it is 1049 times more frequent to find bacterial contamination in PCs compared with HIV, and 1047 times more frequent compared with HCV. Additionally, the frequency of identifying bacterial contamination in PCs is 375 times higher than detecting bacterial infection or sepsis associated with units of red blood cells. Furthermore, it is 232 times more likely to identify bacterial contamination in PCs compared with finding HBV contamination, and 54 times more likely compared with cases of TTBI caused by PCs. However, it is crucial to recognize that TTI rates are highly variable and influenced by various factors, including national testing strategies, the prevalence of specific pathogens, and the implementation of mitigation measures like bacterial testing. Therefore, Figure 3 as presented may not fully represent the exact situation in each country. Careful interpretation is necessary, considering the specific context and contributing factors that could influence the reported rates in different regions.

## 4. Discussion

The administration of blood transfusion is a therapeutic approach commonly utilized in clinical settings [13]. Nevertheless, its application entails not only advantages but also inherent hazards [14]. Despite comprehensive screening measures for HIV, HBV, HCV, and HTLV in all collected blood units, an equal level of screening for the presence of bacteria is not routinely performed in all countries. Multiple surveillance studies focusing on aerobic cultures have revealed that bacterial contamination occurs in approximately 1 in 1000 to 3000 PCs [16]. However, recent data reported by Ramirez-Arcos and Goldman, who studied countries that perform 100% PC blood cultures, indicate that the actual rates of septic-type transfusion reactions are much lower, estimated to be around 1 in 25,000 PCs (with a range of 1 in 13,000 to 100,000) [17]. Over the past two decades, the incidence of RBC transfusion-associated sepsis has significantly decreased, likely due to the wider implementation of leukoreduction filters [16]. Consequently, most post-transfusion sepsis cases observed today are predominantly associated with PCs stored at room temperature.

Taylor et al. conducted a comprehensive study on the impact of temperature on *Burkholderia cepacia* lipid products [33]. Their findings revealed an intriguing array of polar lipids, including two variants each of phosphatidylethanolamine (PE) and ornithine amide lipid (OL). As the growth temperature increased within the range of 25–40 degrees Celsius, there was a corresponding increase in the proportions of molecular species of PE and OL that contained 2-hydroxy acids, while the PE:OL ratio remained unchanged. Remarkably, the balance between neutral and acidic lipids remained unaffected by the growth temperature; however, the contribution of phosphatidylglycerol to the acidic lipids increased as the temperature and growth rate escalated. Furthermore, the pigmentation of cells and the presence of flagella were also found to be dependent on temperature variations.

Figure 4 visually depicts the influence of temperature on the number of generations of *B. cepacia.* Notably, the storage of red blood cell units within the range of 2–6 °C effectively hampers the growth of bacteria despite their presence in the units. However, in the case of PCs, which are stored at temperatures between 20–24 °C, the warm environment provides an ideal breeding ground for the proliferation of this bacteria [34]. Moreover, considering that these units are intended for transfusion to patients whose body temperature typically fluctuates between 36 and 38 °C, it can be inferred that the bacterial inoculum already present in the blood component units will experience favorable conditions for easy proliferation within the patient’s body.

Mariappan et al. demonstrated that *Burkholderia cepacia* exhibits a rapid initiation of growth, characterized by a lag phase of less than two hours. Subsequently, it enters a logarithmic phase, marked by robust proliferation, which typically spans between three and eight hours. Following this phase, the bacterium reaches a stationary phase, indicating a balance between cell division and death, typically occurring after approximately ten hours of cultivation [35]. The duration of the lag phase in bacteria exhibits significant variation, influenced by factors such as species diversity, inoculum size, the bacteria’s life cycle, nutrient availability [36], and the potential bactericidal effect of PCs [37]. Within approximately 3 to 4 days, numerous bacterial groups will attain concentrations exceeding 10^6^, leading to a substantial increase in the initial inoculum upon transfusion. Consequently, this elevated bacterial load significantly raises the likelihood of septic shock development. Notably, three patients reported in Table 1 received PC transfusions collected using apheresis on days 4 and 5 of storage, coinciding precisely with the stationary phase of the Bcc.

In September 2022, the Ramírez-Arcos group made a significant discovery regarding PCs contaminated with bacteria [38]. Their research revealed that not only does bacterial contamination spread within the recipients, but it also impairs the functionality of the platelets. Notably, it was observed that the concentration of glycoprotein IIb, which is crucial for the final phase of platelet aggregation, decreased by 22% after 72 h of contamination and by 40% at five days of contamination. Additionally, the expression of phosphatidylserine, an indicator of premature removal of these essential blood components from circulation, was found to be elevated. These findings imply that transfused PCs contaminated with bacteria are unable to improve primary hemostasis in patients, and they fail to effectively increase platelet counts. Three of the cases described in this review involve PCs and serve as robust examples of this concept, as they did not demonstrate any improvement in post-transfusion platelet counts. This corroborates the notion that contaminated PCs not only lead to infection but also fail to fulfill their intended therapeutic purpose.

The introduction of improved skin disinfection techniques, first aliquot bypass methods, and bacterial detection in PCs has contributed to a decrease in the frequency of clinically significant septic reactions, although they still occur [39]. Some hemovigilance systems found it more effective to conduct culture sampling after 36 h of obtaining the unit [40,41,42], highlighting that timely quality control testing is essential, but it may not detect latent bacteria, leading to false-negative results [36]. Additionally, the study by García-Erce et al. [8] underscores the inherent risk of contamination associated with the manufacturing process of donor arm cleansing products [2]. It also highlights the importance of prompt reporting and rigorous quality control measures to prevent similar incidents in the future.

In many countries, blood banks do not perform bacterial cultures of blood components. For instance, Japan has implemented a PC storage policy, limiting the storage duration to a maximum of 85 h, aimed at reducing the occurrence of septic events in patients [43]. In other countries, blood banks are required to visually inspect all blood components at each stage of processing and before distribution [21]. Any signs of damage, defects, contamination, or abnormalities must prompt the removal of the component [16]. Between 2016 and 2017, a total of 21,808,541 blood components were transfused in eighteen Latin American countries, with red blood cell units accounting for 55.9% and PCs for 20.1% [44]. During the same period, the Pan American Health Organization (PAHO) recorded six cases of transfusion-transmitted bacterial contamination (all reported by Brazil: four in 2016 and two in 2017), resulting in an occurrence rate of one event per 3,634,756 transfused blood components. Usually, these blood banks conduct quality control tests on 1–5% of blood components, checking parameters like volume, cell counts, and microbiological cultures [18]. PCs undergo tests for volume, platelet count, pH, leukocyte count, and microbiological culture [18]. These figures stand in contrast to hemovigilance reports from North America, Europe, Africa, and Oceania, which have reported frequencies of transfusion-transmitted bacterial infections in PCs ranging from 1 in 14,515 to 1 in 384,903, and in red blood cell units ranging from 1 in 96,850 to 1 in 3,448,275 [44]. The contrasting information from these countries highlights the absence of reported cases in the hemovigilance systems of Latin America, excluding Brazil [18].

On the other hand, cases of Bcc pseudo-contamination have also been reported. In 2005, Ebner et al. documented cases of pseudo-contamination involving blood components contaminated with *B. cepacia* during quality controls, stemming from the use of a disinfectant that was found to be contaminated. The disinfectant in question was based on quaternary ammonium, which proved ineffective against certain Gram-negative bacteria, rendering it unsuitable for its intended purpose [45]. Interestingly, all four isolates, including one from the disinfectant and three from the blood components, exhibited identical biochemical reactions and resistance patterns. To address this issue, the authors of the study introduced an alcohol-based preparation, and subsequently, no further instances of *B. cepacia* contamination were identified [45]. Furthermore, reports of Bcc contamination have extended to povidone-iodine solutions [46], highlighting another potential source of concern. In 2007, Vonberg and Gastmeier conducted a comprehensive review of nosocomial infections associated with substance contamination. Their findings revealed that the predominant pathogens in these cases included the hepatitis A virus, *Yersinia enterocolitica*, and *Serratia* spp. specifically associated with blood products. Additionally, *Burkholderia cepacia* and *Enterobacter* spp. were identified as the main culprits in infections related to substances other than blood products [47].

Fadeyi et al. in 2021 describe a case of a lethal septic transfusion response in a 63-year-old individual suffering from chronic kidney and liver disease [48]. The patient had received a pathogen-reduced PC transfusion as a precautionary measure in preparation for surgery. The authors remarked that the specific source of contamination could not be determined. However, it was hypothesized that the contamination possibly occurred due to micro-leaks or imperceptible damage in the matrix bag used for collecting PCs during apheresis, as these defects may have gone unnoticed during manufacturing, transport, storage, processing, or distribution. Notably, Gammon et al. [27] recently demonstrated that certain puncture sites on blood component bags are more susceptible to bacterial contamination, even when using pathogen inactivation techniques.

In 2023, Green et al. conducted a study demonstrating that amotosalen, the only substance approved by the FDA for pathogen inactivation in blood products when combined with ultraviolet light, exhibits structural similarity to substrates of multidrug efflux pumps found in bacteria. These multidrug efflux pumps, known as tripartite resistance, nodulation, and cell division (RND) systems, are prominent in *Enterobacterales*, *Acinetobacter baumannii*, and *Pseudomonas aeruginosa*. They span both the inner and outer membranes of Gram-negative pathogens and actively expel antibiotics from the bacterial cytoplasm into the extracellular space. Notably, *Burkholderia* spp. and other multidrug-resistant bacteria have shown concentrations of amotosalen approaching or exceeding the levels used in approved procedures for plasma and PC inactivation [49]. However, as of now, there has been no investigation conducted to determine whether amotosalen can effectively inactivate the Bcc.

Despite the identification of eleven cases of Bcc in this review, it is highly likely that numerous cases have gone unreported. Therefore, it is crucial for departments of transfusion medicine to empower their staff to actively monitor, diagnose, manage, and report any transfusion-associated reactions encountered. Blood banks and institutions involved in transfusions are strongly encouraged to conduct studies that can address the existing gaps in knowledge. Furthermore, all parties undertaking such studies are urged to share their findings with the national coordination to facilitate potential replication within a country’s specific contexts.

## 5. Conclusions and Recommendations

Our review identified eleven cases of Bcc contamination in blood components, leading to sepsis in the affected patients. These cases were associated with diverse sources, including disinfection solutions used during donor arm cleansing, contaminated water baths, and potential leaks in cryoprecipitate and PC bags. Importantly, the presence of the Bcc in contaminated blood components not only led to infections but also compromised their functionality, hindering their intended therapeutic effect. The significance of implementing rigorous quality control measures throughout the transfusion chain to minimize bacterial contamination risk is underscored by our findings. Additionally, the impact of temperature on Bcc growth emphasizes the importance of appropriate storage conditions for different blood components. The recent in vitro evaluation of amotosalen against various contemporary Gram-negative bacterial isolates has raised concerns about the potential effectiveness of current inactivation methods against multidrug-resistant bacterial pathogens. This necessitates further research to identify potential gaps in safety measures and explore alternative strategies to enhance pathogen inactivation.

Continued hemovigilance practices are of utmost importance to gain a comprehensive understanding of and mitigate the risks associated with TTIs. Strengthening knowledge in this area will contribute to improving transfusion safety and safeguarding patient well-being.

The following recommendations are suggested to prevent cases like those described:Ensure healthcare personnel adhere to hand hygiene and WHO guidelines for blood donation, processing, and transfusion.Provide continuous training and monitoring of personnel in cleaning and disinfection procedures for blood extraction, avoiding palpation after disinfection.Use single-use disposable devices with a mixture of disinfectants for cleaning the puncture area in potential blood donors’ arms.Periodically evaluate cleaning substances used on surfaces to determine their sterility.Ensure biomedical and cold chain equipment used for blood components is regularly cleaned and disinfected to prevent contamination.Reduce PC storage times to less than 85 h.Implement pathogen inactivation technologies before 24 h after collection.Perform culture methods for 100% of PCs after 36 h of storage and before transfusion.Implement rapid detection methods for bacterial contamination prior to transfusion.Conduct detailed and rigorous visual inspections of all blood components and tubing before dispatch and transfusion.Accurately identify non-hemolytic febrile adverse transfusion reactions and consider bacterial contamination as a possible cause.Continuously train healthcare personnel on adverse transfusion reactions to ensure timely identification and management.Encourage reporting of adverse events to hemovigilance programs for analysis and improvement plans to prevent such events.

## Figures and Tables

**Figure 1 microorganisms-12-00040-f001:**
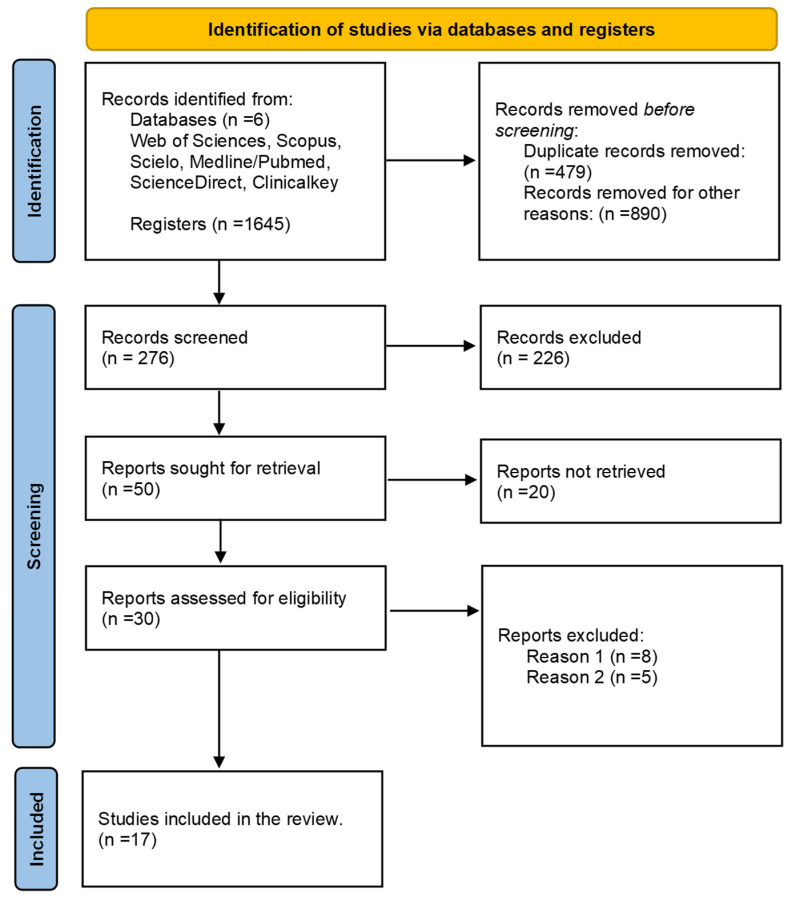
Flow diagram illustrating the selection of studies. From the search conducted, a total of 17 articles were identified that fulfilled the specified characteristics and criteria. These articles were meticulously evaluated using a comprehensive extraction matrix. Reason 1: Infections not associated with the Bcc. Reason 2: Sepsis not related to transfusion of blood components.

**Figure 2 microorganisms-12-00040-f002:**
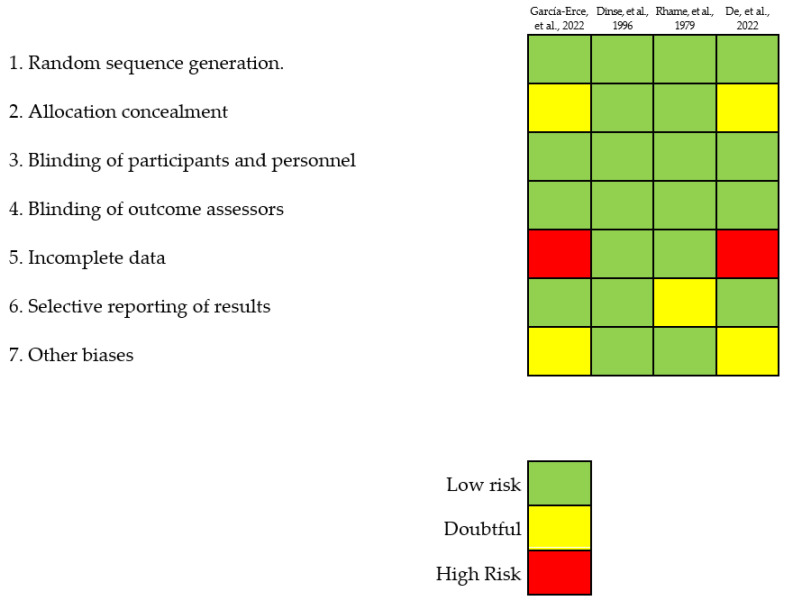
Detailed and disaggregated assessment of the risk of bias for each study [4,8,29,30] following the guidelines set forth by the Cochrane Collaboration.

**Figure 3 microorganisms-12-00040-f003:**
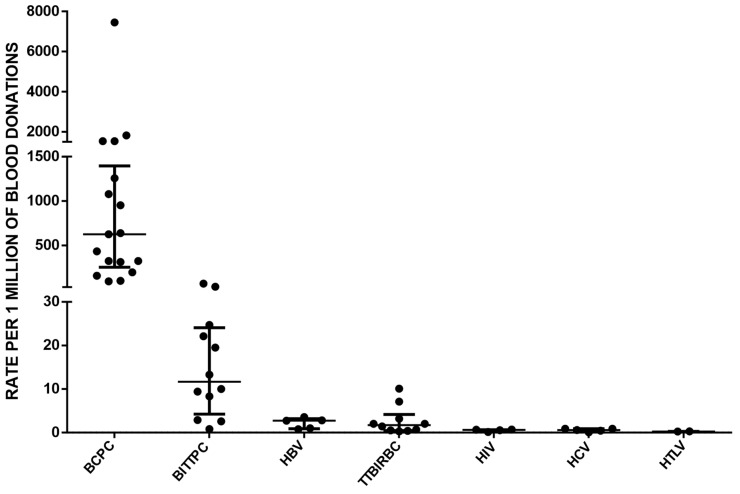
A comprehensive comparison of transfusion-transmitted infection rates was conducted by analyzing hemovigilance reports from multiple countries [16,17,18]. Each point represents the value reported by each country. The abbreviations used in the analysis include BCPC (bacterial contamination in PC), BITTPC (bacterial infection transmitted by transfusion of PC, which includes sepsis cases), HBV (hepatitis B virus), TTBIRBC (transfusion-transmitted bacterial infection in units of red blood cells), HIV (human immunodeficiency virus), HCV (hepatitis C virus), and HTLV (human T-cell lymphotropic virus). The data analysis considered medians and interquartile ranges to provide a comprehensive overview of the transfusion-transmitted infection rates across different pathogens. The medians and corresponding 25th and 75th percentiles were reported to capture the variability in the data and provide a more comprehensive understanding of the infection rates.

**Figure 4 microorganisms-12-00040-f004:**
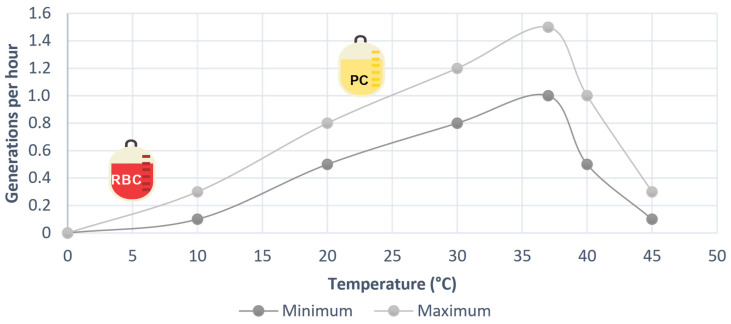
Effect of temperature on the growth rate of *Burkholeria cepacia*, built from data from [34].

**Table 1 microorganisms-12-00040-t001:** Summary of the search strategy.

Database	Search Equation
Medline/PubMed	((“Burkholderia*” [Title/Abstract] OR “cepacia*” [Title/Abstract] OR “Burkholderia cepacia” [Title/Abstract] OR “Pseudomonas cepacia” [Title/Abstract] AND “septic transfusion reaction” [Title/Abstract] OR “transfusion adverse event” [Title/Abstract] OR “transfusion*” [Title/Abstract]))
Web of Science	(“Burkholderia*” or “cepacia*” or “Burkholderia cepacia” or “Pseudomonas cepacia”) AND (“septic transfusion reaction” OR “transfusion adverse event” OR “transfusion*”)
Scopus	(TITLE-ABS-KEY (“Burkholderia*”) AND TITLE-ABS-KEY (“septic transfusion reaction”) OR TITLE-ABS-KEY (“Burkholderia cepacia”) OR TITLE-ABS-KEY (“Pseudomonas cepacia”) OR TITLE-ABS-KEY AND (“transfusion adverse event”) OR TITLE-ABS-KEY (“transfusion*”)
Scielo	“Burkholderia*” or “cepacia*” or “Burkholderia cepacia” or “Pseudomonas cepacia” AND “septic transfusion reaction” OR “transfusion adverse event” OR “transfusion*”
ScienceDirect	“Burkholderia cepacia or “Pseudomonas cepacia” AND “Blood product”
ClinicalKey	“Burkholderia cepacia or “Pseudomonas cepacia” AND “Blood product”

**Table 2 microorganisms-12-00040-t002:** Main characteristics of the patients who developed sepsis due to transfusion of blood components contaminated with the Bcc.

Case	Sex	Age	Primary Diagnosis	Signs and Symptoms during/after Transfusion	Time Elapsed between Transfusion and Onset of Symptoms	Storage Time of the Blood Component before Transfusion	Transfused Blood Component	Microbiological Culture of the Patient	Microbiological Culture of Blood Units	Treatment Post Sepsis Diagnosis	Outcome	Source of Contamination	Imputability of TTBI	Reference
1	Unknown	Unknown	Unknown	Septicemia	Not reported	Not reported	Cryoprecipitate pool	Growth of *B. cepacia*	Not reported	Not reported	Not reported	Serological bath where the cryoprecipitates were thawed	Definitive	[29]
2	Unknown	Unknown	Unknown	Septicemia	Not reported	Not reported	Cryoprecipitate pool	Growth of *B. cepacia*	Not reported	Not reported	Not reported	Serological bath where the cryoprecipitates were thawed	Definitive	[29]
3	Unknown	Unknown	Unknown	Mediastinal wound infection	Not reported	Not reported	Cryoprecipitate pool	Growth of *B. cepacia*	Not reported	Not reported	Not reported	Serological bath where the cryoprecipitates were thawed	Definitive	[29]
4	Female	58	Postoperative vaginal hysterectomy	Fever, signs and symptoms of DIC; progressed to septic shock	Not reported	Not reported	Two red blood cell units	Growth of *B. capacia* and *Serratia marcesens*	Growth of *B. capacia* and *Serratia marcesens*. At 24 h, later growth of *Staphylococcus epidermidis*	Not reported	Not reported	Unknown	Definitive	[30]
5	Male	68	Rectal cancer	Fever, chills, shock, tachycardia, hypotension, stupor; onset immediately after transfusion	Immediately start the transfusion	Not reported	Red blood cell unit	Growth of *B. cepacia*	Growth of *B. cepacia*	Not reported	Not reported	0.5% chlorhexidine aqueous solution	Definitive	[8]
6	Male	28	Acute lymphoblastic leukemia	Fever, tachycardia	Not reported	Not reported	Red blood cell unit	Growth of *B. cepacia*	Not performed	Not reported	Not reported	0.5% chlorhexidine aqueous solution	Probable	[8]
7	Female	59	Gastric bleeding	Fever, chills, shock, tachycardia	Not reported	Not reported	Red blood cell unit	Growth of *B. cepacia*	Not performed	Not reported	Not reported	0.5% chlorhexidine aqueous solution	Probable	[8]
8	Female	*	Myelodysplastic syndrome, arterial hypertension, hypothyroidism	Diaphoresis, dyspnea, tachycardia, tachypnea, fever, septic shock	25 min	4 days	Apheresis PC	Growth of *B. cepacia*	Growth of *B. cepacia*	Cefepime and vancomycin	Hospitalization; she died on 03/10/2021 due to underlying disease	Unknown	Definitive	Institutional Haemovigilance Program. Sociedad de Cirugía de Bogotá-Hospital de San José
9	Male	*	Myelodysplastic syndrome	Diaphoresis, dyspnea, tachycardia, tachypnea, fever, septic shock	25 min	5 days	Apheresis PC	Growth of *B. cepacia*	Growth of *B. cepacia*	Piperacillin tazobactam and vancomycin and daptomycin	Required ICU; he died 12/15/2020 due to underlying disease	Unknown	Definitive	Institutional Haemovigilance Program. Sociedad de Cirugía de Bogotá-Hospital de San José
10	Male	*	myelodysplastic syndrome- hemosiderosis	Severe dyspnea, diaphoresis, tachycardia, fever, tachypnea, ambient desaturation, hypotension, septic shock	35 min	5 days	Apheresis PC	Growth of *B. cepacia*	Growth of *B. cepacia*	Meropenem and vancomycin	Required ICU; recovered from transfusion sepsis but was infected with COVID-19 and passed away 10/27/2020	Unknown	Definitive	Institutional Haemovigilance Program. Sociedad de Cirugía de Bogotá-Hospital de San José
11	Male	57	Acute myeloid leukemia	Septicemia	Not reported	Not reported	PCs and red blood cells	Growth of *B. cepacia* violet pigment	Not reported	Ceftazidime, cotrimoxazole and Imipenem	Not reported	Unknown	Possible	[4]

* Reporting was not feasible as the authors lacked informed consent for the investigation. Unfortunately, the three patients succumbed to their underlying disease between 2020 and 2021, prior to any consideration for inclusion in a study or investigation.

## Data Availability

The data presented in this study are available on request from the corresponding author.

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
