# Peer review of "Septic Transfusion Reactions Involving Burkholderia cepacia Complex: A Review"

_microorganisms, 2023, doi:10.3390/microorganisms12010040_

Round 1

Reviewer 1 Report

Abstract

Line 13: it says “The search encompassed various specialized databases such as Web of Sciences, Scopus, Scielo, among others”, They need to cite the other two

Line 14: it says “An analysis of the literature revealed a total of ten reported cases where blood components contaminated with Bcc had been transfused, resulting in 15 sepsis among the affected patients. Of these cases, seven were documented in the literature, while 16 the remaining three occurred within the institution involving the authors of this document.”

It should be rephrased because as they stated only 7 cases have been reported so “An analysis of the literature revealed a total of ten reported cases” can’t be true.

but there are more cases

De, et al: Unusual violet coloured pigment produced by Burkholderia cepacia complex Unusual Violet Coloured Pigment Journal of Medical Sciences and Health/May-August 2022/Volume 8/Issue 2; 173. The first case described “-History of highgrade fever post chemotherapy - history of platelet and blood cell transfusion -No history of diabetes or hypertension”

Figure 1 should not be in the material and methods section

They have to explain the MM for the systematic review (for example Prysma criteria)

They have to explain the MM for the Quality criteria-àno results (they have to mention the quality results)

The same for the Bias criteria?-àno results

No PICO question in the MM section

Some mistakes such as: Line 186: no bottles are used for transfusion, I think author refers to the donor’s arm disinfection (clarify please)

Line 218: It says “Nevertheless, the isolation of the same pathogen in all three contaminated units of platelet concentrates, as  well as in the transfused patients, suggests that the contamination may have occurred within the bags containing the platelets”. To say so it’s difficult I suppose it has been used the same bag batch from other donors and the problem has not been seen.

There is a mistaken: Pysma diagram should be a result, no quality, no bias results

From line 194 should be discussion (it'snin the results section)

Don’t agree with conclusion

"and it is presumed that the pathogen inactivation process was ineffective due to the intrinsic mechanisms of Burkholderia Cepacia in countering amotosalen."

"To ensure patient safety, it is imperative that blood packs are delivered to blood banks with a sterile exterior and are completely free from moisture."

Please elaborate more (there are more inactivation methods)

Reviewer 2 Report

Summary

The manuscript by Salamanca-Pachon et al reviews the scientific literature on transfusion transmitted bacterial infections of Burkholderia cepacia complex. Twenty different reports for TTBI were evaluated and discussed.

General concept comments

The manuscript is relevant for the field as TTBI of blood products are a cause of serious adverse events with sometimes fatal outcome.

The manuscript however is not very well-structured, making it difficult for the reader to follow the text. The manuscript lacks a concept for the different sections (results, discussion, conclusion).

Specific comments

Title

Page 1, Line 2: Please correct Burkholderia epacia to Burkholderia cepacia 

Introduction:

The introduction contains a lot of information on BCC but only very short part on TTBI.

Please add more information on bacterial risk of blood transfusion e.g. rates of bacterial, contamination, sources of contamination and mitigation strategies.

The part on BCC could be shortened a little bit.

Material and Methods

Search strategy

You limited your search to the title and abstract field of the manuscripts. Classical haemovigilance data (SHOT, haemovigilance reports from FDA) therefore probably was not included in your study. Please discus these limitations in your manuscript.

Figure 1:

Please check the formatting of figure 1.

Results

Table 2: Not all of the cases shown in Table 2 are confirmed transfusion-transmitted bacterial infections. You might add a column to the table with a rating for the evidence of TTBI: e.g. confirmed TTBI, highly likely TTBI, …

It is difficult for the reader to connect the text with the table. I would recommend to use a numbered list for a better correlation of test and table.

The results section contains a lot of discussions on different aspects of bacterial safety and is difficult to follow. Please restrict the results section to the discussion of the transfusion transmitted cases of Burkholderia cepacian complex. General discussions should be moved to the introduction or the discussion section and should be more focused.

Page 9 lines 65ff

The growth of bacteria in blood products has extensively been investigated in in vitro studies. The growth behavior is not only dependent on the bacteria species, but might also be donor-dependent, matrix dependent (PC or RBC) and temperature dependent. Please refer to the respective literature.

Discussions

Page 10 Line 116 ff: Whether or not the Burkholderia cepacian complex can be inactivated by the Amotosalen was not investigated. So please remove these sections indicating that pathogen inactivation of Burkholderia cepacian complex is inefficient.

Figure 3 generalizes transfusion transmitted infection rates very much and it does not take into account, that transfusion transmitted infection rates are highly variable dependent on national testing strategies, prevalences of the respective pathogens, mitigation strategies like bacterial testing e.g.

Conclusion

Please remove the assumption that not inactivated by amotosalen.

The conclusion should be more focused on the mitigation strategies for TTBI in general and Burkholderia cepacia complex in particular.

Author Response

Consulte el archivo adjunto.

Round 2

Reviewer 1 Report

I Think that this version has really improvede the paper

Author Response

thank you. We added the minor corrections requested
